# Developmental Neurotoxicity of Environmentally Relevant Pharmaceuticals and Mixtures Thereof in a Zebrafish Embryo Behavioural Test

**DOI:** 10.3390/ijerph18136717

**Published:** 2021-06-22

**Authors:** Alessandro Atzei, Ingrid Jense, Edwin P. Zwart, Jessica Legradi, Bastiaan J. Venhuis, Leo T.M. van der Ven, Harm J. Heusinkveld, Ellen V.S. Hessel

**Affiliations:** 1National Institute for Public Health and the Environment (RIVM), 3721 AB Bilthoven, The Netherlands; alessandro.atzei@unica.it (A.A.); ingrid-jense@hotmail.com (I.J.); edwin.zwart@rivm.nl (E.P.Z.); bastiaan.venhuis@rivm.nl (B.J.V.); Leo.van.der.Ven@rivm.nl (L.T.M.v.d.V.); ellen.hessel@rivm.nl (E.V.S.H.); 2Environment & Health, VU University Amsterdam, 1081 HV Amsterdam, The Netherlands; jessica.legradi@vu.nl

**Keywords:** developmental neurotoxicity (DNT), psychopharmaceuticals, alternative to in vivo test, zebrafish embryo behavioural test, chemical mixtures, environmental and human risk assessment

## Abstract

Humans are exposed daily to complex mixtures of chemical substances via food intake, inhalation, and dermal contact. Developmental neurotoxicity is an understudied area and entails one of the most complex areas in toxicology. Animal studies for developmental neurotoxicity (DNT) are hardly performed in the context of regular hazard studies, as they are costly and time consuming and provide only limited information as to human relevance. There is a need for a combination of in vitro and in silico tests for the assessment of chemically induced DNT in humans. The zebrafish (*Danio rerio*) embryo (ZFE) provides a powerful model to study DNT because it shows fast neurodevelopment with a large resemblance to the higher vertebrate, including the human system. One of the suitable readouts for DNT testing in the zebrafish is neurobehaviour (stimulus-provoked locomotion) since this provides integrated information on the functionality and status of the entire nervous system of the embryo. In the current study, environmentally relevant pharmaceuticals and their mixtures were investigated using the zebrafish light-dark transition test. Zebrafish embryos were exposed to three neuroactive compounds of concern, carbamazepine (CBZ), fluoxetine (FLX), and venlafaxine (VNX), as well as their main metabolites, carbamazepine 10,11-epoxide (CBZ 10,11E), norfluoxetine (norFLX), and desvenlafaxine (desVNX). All the studied compounds, except CBZ 10,11E, dose-dependently inhibited zebrafish locomotor activity, providing a distinct behavioural phenotype. Mixture experiments with these pharmaceuticals identified that dose addition was confirmed for all the studied binary mixtures (CBZ-FLX, CBZ-VNX, and VNX-FLX), thereby supporting the zebrafish embryo as a model for studying the cumulative effect of chemical mixtures in DNT. This study shows that pharmaceuticals and a mixture thereof affect locomotor activity in zebrafish. The test is directly applicable in environmental risk assessment; however, further studies are required to assess the relevance of these findings for developmental neurotoxicity in humans.

## 1. Introduction

Embryonal and foetal development of the central nervous system is complex and occurs in strictly controlled timeframes, involving many different processes at the molecular, cellular, and tissue levels, such as cell proliferation, differentiation, migration, axon guidance, and network formation [1,2]. Perturbation of these processes by genetic and environmental factors, such as chemical exposure, might cause neurodevelopmental disorders, including attention-deficit/hyperactivity disorder (ADHD), autism, learning disabilities, intellectual disabilities (also known as mental retardation), conduct disorders, and impairments in vision and hearing [3]. The exact causes of these disorders are currently not clear. Due to the still immature blood–brain barrier (BBB) and reduced ability to detoxify exogenous chemicals, the developing nervous system is more vulnerable to the neurotoxic effects of chemicals than the adult nervous system [1,4]. Systematic testing of DNT is not mandatory in international regulations for admission of pharmaceuticals or industrial chemicals. For regulatory intents, the detection of chemicals with DNT potential is mainly based on developmental in vivo studies in rats, i.e., the OECD-TG-426 or the DNT cohort of the OECD-TG-443, although actual testing for DNT-related effects of chemical exposure in these TGs occurs only if there is a trigger for DNT in other obligatory regulatory tests [5,6]. However, the predictivity of these animal tests for human health effects is uncertain given model differences and because of the relatively nonspecific or insensitive endpoints that are generally used to investigate DNT in animals [7]. Furthermore, such tests are expensive and time consuming and therefore unsuited to screen large numbers of chemicals. This highlights the pressing need to develop alternative in vitro and in silico test methods, preferably considering human-relevant mechanistic data on DNT, and preferably integrated into innovative testing strategies to predict DNT, as previously explained in general terms [6]. The major in vitro methods applied are stem cell-based methods, including (human) induced pluripotent stem cells, using neuronal cell relevant endpoints, and several nonmammalian embryonal models, of which the zebrafish embryo is the most studied model [8]. In silico methods, such as QSARs, read across, computational modelling, etc., generally have good potential for screening and prioritizing toxicants, although only QSARs have been studied in the context of DNT so far [8]. Such strategies should consider the complex anatomy of the human brain, as compared to other vertebrates. An initiative to structure toxicity data, including mechanisms of toxicity, is through the construction of adverse outcome pathways (AOPs) [9], which can also be used to describe the pathways leading to DNT and to select relevant in vitro tests to measure key events in such an AOP for DNT [10,11]. In the context of AOPs, zebrafish provide the much-needed bridge between (complex) cell models and higher organisms (being an in vitro whole-organism model), ultimately allowing for a better interpretation of the consistency and translatability throughout the AOP or AOP network.

Zebrafish (*Danio rerio)* are increasingly recognized as a valuable model for studying chemical-induced toxicity, not only in the field of environmental toxicology but also in human toxicology. There are several reasons why zebrafish provide a potentially powerful model for DNT testing. This includes practical reasons, such as external fertilization, high reproduction rates, and small-sized and transparent embryos allowing for direct observation of developmental delay and malformations. Additionally, embryonic zebrafish (ZFE) up to 120 h postfertilization (hpf) are not considered as experimental animals under European legislation. As such, it is one of the limited new approaches to whole-organism test systems that includes the development of a whole brain. Moreover, the specific applicability of the ZFE for DNT testing is related to the high level of evolutionary conservation of processes involved in the development of the brain, the presence of most (human-relevant) neurotransmitter pathways, and the development of a blood–brain barrier (BBB) [12]. Carbamazepine (CBZ), fluoxetine (FLX), and venlafaxine (VNX) are psychoactive pharmaceuticals acting on the central nervous system (CNS) [13,14]. CBZ is a sodium channel blocker used in the treatment of epilepsy, bipolar disorders, and neuralgia [15]. FLX is a selective serotonin reuptake inhibitor (SSRI) used for the treatment of depression, anxiety, compulsive behaviour, and eating disorders [16]. VNX is a serotonin norepinephrine reuptake inhibitor (SNRI), also prescribed for the treatment of depression [15]. These drugs can cross the placental barrier and therefore may reach the embryo [17,18,19]. FLX and VNX are classified as US pregnancy category C, indicating possible human risks in pregnancy based on animal experiments, which should be balanced against the specific therapeutic benefits. CBZ is classified as category D: there is positive evidence of human foetal risk based on studies in animals or humans, but potential benefits may warrant the use of the drug in pregnant women despite potential risks. Whether the drugs cause teratogenic effects is not conclusive, but studies show some abnormalities of the cardiovascular system for FLX and VNX [20,21,22], and some major congenital malformations are reported for CBZ [23,24]. Whether these drugs cause an adverse effect on the developing nervous system at therapeutic levels requires further investigation [25,26], but their pharmaceutical mode of action is relevant for normal brain development, and disruptions might contribute to the neurodevelopmental disorders. For CBZ, FLX, and VNX, effects in aquatic organisms, including zebrafish, have been reported. For CBZ, it has been shown that it alters the zebrafish’s behaviour in conjunction with histopathological changes in the brain, and lethality is observed at high concentrations [27,28,29]. For FLX, it has been shown that it alters zebrafish behaviour, resulting, for instance, in reduced anxiety-related behaviour [30,31]. For VNX, recent studies have shown that VNX exposure starting at early stages can alter neurobehaviour at 5 dpf [32,33]. In addition, for FLX and VNX transgenerational changes in neurobehaviour have also been reported [34,35,36].

Pharmaceutical residues are present in surface and drinking water in the Netherlands [37]. According to an estimation made by the National Institute for Public health (RIVM), 140 tons of pharmaceutical contaminants (including their metabolites) and 30 tons of radiographic contrast agents end up in Dutch surface waters every year [38] through human or animal excrements. Undegradable pharmaceutical residues mix with surface water and may subsequently end up in drinking water [37]. Although the current concentrations are not considered a threat to the quality of drinking water, a risk for human DNT may still exist in view of a lack of a valid testing strategy for DNT effects. An American study linked the presence of these psychoactive pharmaceuticals in the environment to neurodevelopmental toxicity in another fish species, *Pimephales promenas* (fathead minnow), assessed in adult fish [15,39]. The aim of our study is to test whether these DNT effects can be reproduced in the zebrafish embryos, thereby assessing whether this ZFE model can have a role in DNT screening for psychoactive pharmaceuticals acting on the development of the central nervous system (CNS) [13,14]. The extrapolation of effects in ZFE to humans is discussed, in addition to an assessment of the impact on environmental species, in view of the direct translatability of such effects to environmental species. Therefore, the same pharmaceuticals, their main metabolites, and binary mixtures of the drugs were tested in the light-dark transition test. To assess the DNT effects of these pharmaceuticals, we studied their embryotoxicity, induction of (irreversible) effects on neurobehavior, and effects on gene expression of specific markers related to neurotoxicity.

## 2. Materials and Methods

### 2.1. Chemicals

The test compounds carbamazepine (CBZ, CAS 298-46-4; cat. no. C4024), carbamazepine 10,11-epoxide (CBZ10,11E, CAS 36507-30-9; cat. no. C4206), fluoxetine hydrochloride (FLX, CAS 54910-89-3; cat. no. F132), norfluoxetine hydrochloride (norFLX, CAS 57226-68-3; cat. no. F133), phenytoin (PHT, CAS 57-41-0, cat. no. P1290000), venlafaxine hydrochloride (VNX, CAS 93413-69-5; cat. no. V7264), and desvenlafaxine hydrochloride (desVNX, CAS 300827-87-6; cat. no. D2069) were all obtained from Sigma-Aldrich (St. Louis, MO, USA). VNX, FLX, and their metabolites were diluted directly in embryo medium (see below). CBZ, its metabolites, and PHT were dissolved in dimethyl sulfoxide (DMSO; Merck, Darmstadt, Germany) and further diluted in embryo medium, with a final concentration of 0.1% DMSO.

### 2.2. Maintenance of Fish and Egg Spawning

Experiments with zebrafish (*Danio rerio*) were performed at two locations, i.e., at the zebrafish facilities of National Institute for Public Health and the Environment (RIVM), Bilthoven, and Free University Amsterdam (VU), both using a breeding line of fish which were originally obtained as a commercial wild-type import (Ruinemans Aquarium BV, Monfoort, The Netherlands); in addition, a breed of AB-line zebrafish obtained from the European Zebrafish Resource Centre (Karlsruhe, Germany) was used at RIVM to replicate some of the single-compound experiments and to assess the robustness of the effects. At RIVM, fish were kept and bred using 7.5 L ZebTec tanks (Tecniplast S.p.A, Buguggiate, Italy), with a photoperiod of 14/10 h light/dark (gradual on and off turning), temperature maintained at 27.5 ± 1 °C, pH at 7.5 ± 0.5, and conductivity at 500 ± 100 μS. Fish were fed twice a day with SDS 100, 200, 400, or small granules (Special Diet Services, Essex, UK), depending on the age of the fish, and supplemented with *Artemia salina* (three times per day in-house cultured live artemia for larvae and young juveniles; defrosted artemia obtained from Ruto Frozen Fish Food Zevenhuizen, The Netherlands, once daily for adults). To obtain embryos for the experiments, females were separated from males four days prior to spawning and fed artemia three times/day. The afternoon before spawning, two females and two males were reunited as breeding units in breeding tanks, and spawning was initiated by dawn.

### 2.3. Zebrafish Embryotoxicity Test (ZFET)

The zebrafish embryotoxicity test (ZFET [40]) was applied to determine the embryotoxicity potency of the test compounds in fish. Spawned eggs were collected with a sieve and rinsed thoroughly with embryo medium (demineralized water supplemented with 100 mg/L NaHCO_3_, 20 mg/L KHCO_3_, 200 mg/L CaCl_2_·2H2O, and 180 mg/L MgSO_4_·7H2O) and quality was checked under a microscope. Batches with less than 10% coagulated eggs and limited egg deformations were pooled. Eggs at 4–32-cell stage were selected within 2.5 h postfertilization (hpf) and transferred to a 6-well plate (10 eggs per well) containing 5 mL of test medium with a dilution range of each test compound in embryo medium, including maximum dissolution as the highest concentrations (Appendix A), each with appropriate blank controls (0.1% DMSO for CBZ, its metabolites, and PHT; plain embryo medium for FLX, VNX, and their metabolites). Immediately after selection, the eggs were placed into a 24-well plate (1 egg in 2 mL per well). The 24-well plates were kept in an incubator at 27.5 ± 1 °C with a light/dark cycle of 14/10 h. After 3 days post-fertilization (dpf), the developmental and teratological effects of the embryos were evaluated under a light microscope as described previously [40]. In brief, development was scored using an integrative semiquantitative scoring system (general morphology score, GMS) for specific developmental endpoints, including detachment of tail, formation of somites, development of eyes, movement, heartbeat, blood circulation, pigmentation of head/body, pigmentation of tail, pectoral fin, protruding mouth, and hatching. In addition, teratological effects were scored as present or absent as a total teratology score, considering pericardial oedema, yolk sac oedema, eye oedema, malformation of the head, absence/malformation of sacculi/otoliths, malformation of tail, malformation of heart, modified chorda structure, scoliosis, rachischisis, and yolk deformation.

### 2.4. Light-Dark Transition Test

The light-dark transition test measures locomotory behaviour under light and dark conditions. This readout was used as a phenotypic marker for (developmental) neurotoxicity, and the effects hereon were studied by measuring the activity of 5-dpf embryos after continuous exposure to the target compounds, in a dose-range enabling assessment of effects induced at environmental levels. Although locomotor activity may be affected by factors other than neurological effects, it is a sensitive endpoint for DNT assessment because it depends on the integrity of brain function, nervous system development, and visual pathways, and the endpoint can therefore be used to screen for DNT effects of chemicals [41]. Exposure to single compounds was started within the first 2.5 hpf and was terminated at 5 dpf with the evaluation of swimming activity. For that purpose, fertilized eggs were firstly exposed in a 6-well plate (20 eggs per concentration and solvent control) containing 5 mL of test medium and kept in an incubator at 27.5 ± 0.5 °C up to 5 dpf. Before performing the behaviour test, embryos were moved along with 300 µL of test medium to a 96-well plate (1 embryo per well) for a total of twelve (*n = 12*) embryos per concentration. At 120 hpf, after acclimatization for 30 min in light, free swimming activity was recorded in the ZebraBox (Viewpoint, Lyon, France) during three repeated triggers of light-dark transitions in 10-min periods. Sensitivity was set at 20, and thresholds were 10 (burst) and 1 (freezing). Locomotor activity was evaluated as the total duration of movement per 10 min, using the Zebralab Quantization software (Viewpoint, Lyon, France), which gives “time in activity” as output. Occasionally occurring embryos with observable morphological aberrations were excluded from behaviour testing to avoid obvious non-neurological causes for observed effects on locomotor activity [42]. For the first set of experiments, behaviour was tested in a dose–response setup at concentrations below visually observable embryotoxicity in the ZFET or at the dissolution limit in cases where no embryotoxicity was observed. Half-logarithmic dilutions were applied as shown in Table 1. These experiments were replicated at the two locations VU and RIVM. Environmentally relevant concentrations were tested and repeated for the parent compounds at concentration ranges reported in Table 1. The sensitive window of exposure and (ir)reversibility of effects were studied using a single effective concentration for each compound (nominal concentrations 200, 10, and 300 µM for, respectively, CBZ, FLX, and VNX), with short, defined exposure windows (<2.5–96 hpf, 96–120 hpf), as compared to the full period of exposure (<2.5–120 hpf).

### 2.5. Mixture Design

Zebrafish embryos were exposed to the binary mixtures of carbamazepine-fluoxetine (CBZ-FLX), carbamazepine-venlafaxine (CBZ-VNX), and venlafaxine-fluoxetine (VNX-FLX). For each mixture, the concentrations of the second compound B were expressed as equivalents of the first (reference) compound A, thus adjusting for the difference in potency using a relative potency factor (RPF) [43]. CBZ was the reference compound in its combinations with FLX and VNX, while VNX was the reference compound of the VNX-FLX mixture. The RPFs were calculated using a dedicated function in the PROAST software (RIVM, Bilthoven, The Netherlands) (see below) and/or by comparing the BMC_50_ of the two compounds (Table 2). The resulting RPFs enabled an equipotent dose range of the mixtures to be designed, aiming to cover the intermediate part of the single dose–response curve of the reference compound. In addition to the 1:1 ratio of equipotency, the excess ratios 1:3 and 3:1 were investigated to account for potency/sensitivity variations between experiments. Exposure to mixtures was performed as described for the single-compound analyses and always accompanied by single-compound doses to account for interexperimental potency variations. Behavioural tracking was performed at 120 hpf as described above.

### 2.6. RNA Isolation and Quantitative Real-Time PCR

Specific gene expression markers related to DNT were derived from a previous study in fathead minnows [15] and included: *gabra6a* (Genbank: NM_200731.1), *grin1a* (Genbank: NM_001076714.2), and *dlg4* (Genbank: NM_214728.1) (Applied Biosystems). As the negative control, the following housekeeping genes were used: *gapdh* (Genbank: NM_001115114.1), *actb1* (Genbank: NM_131031.1), and *hprt1* (Genbank: NM_212986.1). All targets were obtained as standard assays (Applied Biosystems). Gene expression was measured at BMC_50_ values, i.e., the benchmark dose where 50% of the locomotor activity was inhibited, as observed in a previous experiment with matched conditions and based on pooled data from three subsequent dark blocks. This analysis produced the following BMD_50_ values: 115 μM for CBZ, 6 μM for FLX, and 107 μM for VNX. Six replicate pools per condition, each containing 10–12 embryos, were exposed during 0–120 hpf, then euthanized in liquid nitrogen and stored at −80 °C. For RNA isolation, the RNeasy Mini kit (QIAGEN, Venlo, The Netherlands) was used according to the manufacturer’s protocol. Briefly, frozen embryos were pulverized using a tissue homogenizer (Omni TH) in a 2-mL Eppendorf tube, lysed in Qiazol and chloroform, and centrifuged. The aqueous phase was removed, mixed with EtOH (70%) and RNA was extracted using the dedicated RNeasy column. The concentration of RNA was measured on the NanoDrop spectrophotometer 2000 c (Thermo Fisher Scientific, Waltham MA, USA) as A260/A280 and A260/A280 ratios, and RNA integrity was assessed on the Bioanalyzer 2100 (Agilent, Waldbronn, Germany) using the RNA 6000 Nano Chip kit (Agilent). Samples with a NanoDrop score ≥ 1.8–2.0 and an RNA Integrity Number (RIN) between 7 and 10 were considered of sufficient quality for further qPCR analysis. The samples of isolated RNA were stored at −80 °C. For qPCR, RNA was transcribed to complementary DNA (cDNA) with the High-Capacity cDNA Reverse Transcription Kit (Applied Biosystems, Fisher Scientific, Landsmeer, the Netherlands) according to the manufacturer’s protocol. The target genes were amplified during qPCR with the Applied Biosystems 7500 fast real-time PCR system with software v2.0.6 (Thermo Fisher Scientific, Waltham, MA, USA).

### 2.7. Statistical Analysis

#### 2.7.1. ZFET—Single Compounds

The morphology and teratology scores obtained from ZFET as well data from behaviour testing were used to perform a benchmark dose–response analysis with PROAST software v67.0-70.0 (RIVM, Bilthoven, The Netherlands) (https://www.rivm.nl/en/proast/, accessed 26-05-2021 [44]), as a package in R statistical software v3.6.0-4.0.0 (RIVM, Bilthoven, The Netherlands). PROAST is also available as a web application (https://proastweb.rivm.nl/, accessed 26-05-2021) and as an integrated part in the EuroMix toolbox (https://mcra-test.rivm.nl/EuroMix/WebApp/#/, accessed 26-05-2021). The dose–response analysis enables the estimation of a benchmark concentration (BMC) at a defined critical effect size (CES). A BMC_05_ (BMC at CES = 5%) was derived for the ZFET, and a BMC_05_ and BMC_50_ for behaviour testing. The estimated BMC is reported along with its lower (BMCL) and upper (BMCU) bound at its 90% confidence interval.

#### 2.7.2. Light-Dark Transition Test—Single Compounds

All the dose–response analyses were repeated with exponential and Hill models. The data of the different exposure windows are presented as geometric mean with 90% confidence intervals of total time in activity per 10-min measurement of the first dark block only of *n* replicate embryos. The second and third dark blocks did not provide additional information (see Appendix A), whereas the light blocks did not show any statistical differences (see Appendix A).

#### 2.7.3. Gene Expression—Single Compounds

Relative expression levels were calculated using the 2^−ΔΔCT^ method [45], which considers the expression of each marker gene compared to the mean expression of the three reference genes and the compound-induced expression compared to the background expression level in blank controls. The resulting fold change values were expressed as log2FC. Student’s t-tests (unpaired, two-tailed) were performed to determine significance (*p* < 0.05).

#### 2.7.4. Mixtures and Light-Dark Transition Test

The PROAST software was also used to calculate the relative potency factor (RPF), combining the data of the refence and second compound dose–response analysis. The evaluation of the dose addition was performed both in a visual and quantitative way. The first way was applied by visual comparison of the dose–response curve fitted to both the single-compound responses and the mixture responses, after expressing all concentrations in equivalent units of the reference compound after transformation using the RPF. Dose addition is likely when all data (mixture and single-compound) are described by the fitted curve. The visual assessment was supported by a quantitative evaluation that consisted of a comparison between the RPFs-CIs calculated with and without mixture data. When the dose addition holds, an overlap of the RPFs-CIs is expected (RPF does not change when including the mixture in the analysis); this can be quantified by dividing the RPFL (relative potency factor lower) of the higher interval to the RPFU (relative potency factor upper) of the lower interval. Ratios greater than 1 indicate a relatively large deviation from dose addition, while a ratio smaller than 1 means that there is no evidence of deviation from dose addition. 

## 3. Results

### 3.1. Zebrafish Embryotoxicity Test (ZFET)

After exposing the zebrafish embryos to the test compounds up to 72 hpf, only FLX and its metabolite norFLX resulted in a concentration-dependent decrease in GMS, with a BMC_05_ of 31.2 (CI 7.22–57.2) µM and 32.19 (CI 14.9–59.7) µM (Appendix A). Particularly, a high mortality rate of the embryos was observed when exposed to the highest tested concentrations of FLX and norFLX, 89.9 and 60 µM, respectively. No developmental delay was observed with the other compounds, and no teratogenicity was detected.

### 3.2. Behaviour Testing

#### 3.2.1. Single-Compound Dose–Response Analysis

As a first step in locomotor analysis, the locomotor activity was evaluated after exposure to the high range of laboratory concentrations (Table 1). Here, all test compounds, except CBZ10,11E, showed a dose-dependent decrease in the dark period of the embryo locomotor activity. The example output of the quantization protocol with CBZ (Figure 1A) illustrates the multitude of parameters which can be analysed, including light-dark transition values, initial and peak values, change over the three measurement blocks, etc. This report further considers the analysis of the first 10-min dark block only; in this study, the second and third blocks did not provide additional information, and there were no significant effects in light blocks (see Appendix A, Appendix A). The total duration of activity in those 10-min sections was recorded per embryo and analysed for dose responses in dark and light (Figure 1B). Given the absence of an effect during light with each compound, this parameter was not considered to be of informative use and not further reported. In this way, the independent replicate experiments of CBZ, FLX, and VNX displayed a strong reproducibility between the two different laboratories, resulting in a graphical and quantitative overlap of each replicate dose–response curve (Figure 2). This highlights the robustness of the effects and the reproducibility of the test, especially since each laboratory uses their own strain. The quantization output for FLX, PHT, and VNX and graphical representation of CI overlap are reported in the Appendix A. When summarizing the results of the single-compound dose–response analysis (Table 3), it appeared that FLX and VNX had a similar potency (BMC_05_ ranges of 0.17–0.65 and 0.26–1.4µM, respectively), and that CBZ and PHT were about 85x less potent compared to these two compounds (BMC_05_-CBZ range 51.6–82.3µM; BMC_05_-PHT 45.13µM). The metabolites were either ineffective (CBZ10,11E) or showed a lower potency than their respective parent compounds (BMC_05_ around 2 and 6 µM for norFLX and desVNX, respectively). Relative potency analysis was calculated using the same data sets, although at CES = 50%, for the design of the subsequent mixture experiments as an alternative to a similar RPF analysis using PROAST. These values (Table 3) revealed FLX as the most potent compound (BMC_50_ average around 3µM), followed by an approximately 17x and 50x lower potency of VNX and CBZ, respectively. Exposure to the parent compounds, CBZ, FLX, and VNX, was repeated at environmentally relevant concentrations to confirm the absence of effects at these levels (Table 1). Indeed, no effect was observed at both light and dark periods (not shown).

#### 3.2.2. Different Exposure Windows

Different exposure periods were investigated to assess the persistence of pharmaceutical-induced effects on zebrafish locomotor activity. Embryos were exposed to the effective concentrations of CBZ, FLX, and VNX (200, 10, and 300 µM, respectively) during 0–120, 0–96, and 96–120 hpf time intervals. In all conditions, the exposures already induced a decreased embryo activity at 96 hpf over the dark periods (*p* < 0.05) (Figure 3), although the inhibition with FLX appeared less pronounced than with the other two compounds. Removal of the pharmaceuticals at 96 hpf led to a motor activity recovery at 120 hpf with CBZ and FLX, but not with VNX (Figure 3). Furthermore, the acute exposure (96–120 hpf) also induced a decreased activity compared to the control (*p* < 0.05) and was almost as effective as the chronic treatment (0–120 hpf) with all three compounds.

#### 3.2.3. Gene Expression

To better investigate the possible developmental neurotoxicity mechanism induced by CBZ, FLX, and VNX in zebrafish embryos, the gene expression of the specific DNT markers, *grin1a*, *dlg4*, and *gabra6a,* was studied. The mRNA expression of all three target markers was upregulated with exposure to 115 µM CBZ (*p* < 0.05), although most markedly for *dlg4*, which reached a double relative quantification compared to the control (Figure 4). In contrast, both exposures to 10 µM FLX and 107 µM VNX showed a significant upregulation of *gabra6a* mRNA expression only (*p* < 0.05).

#### 3.2.4. RPF Estimation for Mixture Design

To design accurate mixture experiments, an RPF was estimated for each binary combination of the test compounds, based on their individual potencies (expressed as BMC_50_). A combined dataset including the first dark periods of each compound was analysed in a single run by PROAST for CBZ-FLX and VNX-FLX mixtures, whereas, in the case of the CBZ-VNX mixture, a manual comparison was preferred because of their different DR curve shapes. This analysis revealed that FLX was 50× and 10× more potent than CBZ and VNX, respectively, whereas the latter was 12x more potent than CBZ (Table 2).

#### 3.2.5. Mixture Results

The binary mixtures of CBZ-FLX, CBZ-VNX, and VNX-FLX, together with the exposure to single compounds, were graphically and quantitatively evaluated in order to assess the dose-addition model in predicting the combined effect of the chemical mixtures. The mixtures’ dose–response curves (Figure 5) describe the behaviour of the single compounds along with the three compound combinations in equipotent (1:1) and near-equipotent (1:3; 3:1) ratios. The visual evaluation shows that the responses of both single compounds and mixtures do not deviate from the fitted curve describing the trend of the entire pool of data. In the event of deviation from dose addition, the graphical assessment would show a shifting of the mixture response either to the right (less than dose addition) or to the left (more than dose addition). A quantitative evaluation objectively supported the graphic estimation by comparing the RPF-CI calculated with and without mixture (Table 4). Indeed, a ratio overlap below 1 was obtained for all the studied binary mixtures, supporting that the mixture data did not affect the RPF, which is in line with dose addition. This conclusion is true for the combination of compounds with similar (VNX-FLX) and dissimilar modes of action (CBZ-FLX and CBZ-VNX).

## 4. Discussion

### 4.1. Prenatal Exposure to Psychoactive Compounds May Lead to Long-Term Neurobehavioural Outcomes

Neurodevelopmental disorders (NDDs) show an increasing incidence worldwide during the last two decades, as indicated by epidemiologic studies [46,47,48]. NDDs represent a major cause of lifelong, chronic impairment marked by difficulties in personal, social, educational, or occupational functions, in addition to having a strong impact on the life quality of entire families [49]. Human studies show associations with the occurrence of NDDs with exposure to chemicals in general. However, only just over ten substances, including some metals (lead, manganese, methylmercury), inorganic compounds (polybrominated diphenyl ethers, fluoride, arsenic, polychlorinated biphenyls), organic solvents (toluene, dichlorodiphenyltrichloroethane, tetrachloroethylene), and pesticides (chlorpyrifos), are currently considered as the most important developmental neurotoxicants [50,51]. Embryonal exposure to chemicals might perturb the complex CNS development leading to long-term brain damage. In this study, the applicability of the zebrafish light-dark transition test as one of the models to screen compounds for DNT was investigated by exposing zebrafish embryos to three psychoactive pharmaceuticals, carbamazepine (CBZ), fluoxetine (FLX), and venlafaxine (VNX), as well as their main metabolites, carbamazepine-10,11-epoxide (CBZ-10,11-E), norfluoxetine (norFLX), and desvenlafaxine (desVNX). This study further aimed to assess their potential (developmental) neurotoxicity as single substances and as binary combinations.

### 4.2. The Target Compounds Inhibit the Locomotor Activity of Zebrafish Embryos

All the studied compounds, except for CBZ-10,11-E, appeared to induce an inhibitory effect on the zebrafish embryo locomotor activity. These outcomes are supported by other comparable studies, both in terms of apical effect and concentrations used. For instance, 5–6-dpf zebrafish embryos exposed to ≥ 180 µM CBZ [27], about 5 µM FLX [31], and 11 µM VNX [52] displayed a significant decrease in swimming activity, which suits the BMC_50_ values observed in our study (Table 3). However, different effective doses are also reported. CBZ doses ranging from 0.04 to 0.85 µM decreased the mean swimming speed of *Jenynsia multidentate* fish compared to the control [53], whereas roughly 26 µM of CBZ reduced the swimming speed in medaka fish (*Oryzias latipes*) [54]. Additionally, FLX concentrations of ≥ 0.003 µM [55,56] seemed to decrease the swimming activity of 5–7-dpf zebrafish, and 0.32 µM of VNX strongly reduced the motility in of 6-dpf zebrafish [56]. Such differences may be mostly explained through variations in experimental design among these studies [53,54,55,56], which is supported by overlapping results when experimental parameters were similar (i.e., same study model or comparable exposure regime and measured endpoint) [27,31,52]. The observed differences support that harmonization of the assay as well as understanding of the sensitive windows in zebrafish neuronal development is imperative when intended to apply for regulatory purposes.

### 4.3. MOA of the Target Compound, Potential Relation to DNT

The inhibition of locomotion caused by the tested pharmaceuticals could be explained by a general toxicity effect or by the specific mode of action of the target compounds. In this respect, CBZ is a voltage-gated sodium channel blocker, while additional actions include GABA (gamma-aminobutyric acid) agonism and increasing the concentration of serotonin at neuronal synapses [57]. In the developing zebrafish, the GABAergic nervous system is one of the first to be established and it plays an important role during early brain development [58]. CBZ can increase the activity of the GABA receptor, thereby stimulating the inhibitory effect of GABA on the CNS [59,60]. The inhibitory effect may also decrease the action potential transduction by blocking the voltage-gated sodium channels. This effect appears at specific threshold values (millivolt); hence, it might cause a sudden collapse of the zebrafish motor activity. This seems reflected in the CBZ dose–response curve characterized by the maximal steepness value set in the analysis (d = 4; Figure 1B), which could reflect the achievement of the threshold limit required to close the sodium channels. The dose–response curve of the similarly acting drug phenytoin (PHT) showed the same high steepness (d = 4, Appendix A), supporting that the key MOA of antiepileptic drugs in zebrafish motor behaviour modulation may be the voltage-gated sodium channel bushing. FLX is an SSRI; therefore, the inhibitory locomotor effect may be due to the serotonergic modulation that in zebrafish embryos starts at 4 dpf [31]. The accumulation of serotonin in the synaptic cleft caused by the pharmacological inhibition of its reuptake can overstimulate the 5-HT receptors, resulting in a downregulation of these receptors [13]. Since serotonin plays an important role in modulating the motor output, the alteration of the serotonergic system may disrupt motor activity control [61]. This mechanism is supported by the observed decrease of locomotor activity in zebrafish embryos, which correlated with a reduction in two serotonin receptor transcripts (SERT and 5-HT1A) in the spinal cord after exposure to FLX [31]. Alternatively, FLX-mediated induction of neurosteroid production (e.g., allopregnanolone) has been proposed as a mechanism of (pharmaceutical) action for anxiolytics in mammals [62,63]. Indeed, it has been confirmed that several neuropeptides associated with stress and anxiety are being regulated in the case of FLX exposure [30].

VNX is an SNRI with slightly different therapeutic uses than FLX. The pharmacological mechanism of action of VNX revolves around the modulation of serotonergic as well as norepinephrine and dopamine reuptake. This mechanism is shown to be responsible for the decreased behavioural responses as a result of VNX exposure in fish [32,33,64,65,66] and may be related to enhanced neurogenesis in the hypothalamus, dorsal thalamus, and preoptic area [32]. This is considered to underlie the persistent developmental neurotoxicity of VNX. Nevertheless, for both VNX and FLX, additional sex-specific transgenerational effects on neurobehaviour have also been demonstrated, rooted in the disruption of the cortisol stress axis [34,35,36]. The lack of locomotor recovery observed in 5-dpf zebrafish exposure to VNX, indeed, supports a DNT involvement for this compound. On the other hand, the removal of pharmaceuticals at 4 dpf led to the recovery of locomotor function in 5-dpf zebrafish exposed to CBZ and FLX. This either indicates no persistent interference of these two compounds with the development of the CNS or indicates that developmental stages up to four days do not present a sensitive window for CBZ- and FLX-induced DNT. Indeed, in the literature, it is described that the exposure to FLX between 4 and 6 dpf induced a significant decrease in swimming activity, persistent up to 14 dpf, whereas this effect was not observed with exposure before 4 dpf [31]. This suggests that 4–6 dpf represents a critical period for the development of spontaneous swimming activity, which should therefore be further explored when analysing the persistence of locomotor effects. To summarize, while the locomotor inhibitory effect may be expressed similarly across different studies, species, and compounds, the underlying modes of action may be different and thereby affect sensitive windows to induce the persistence of effects and thus to induce DNT.

### 4.4. The Contribution of Gene Expression to DNT Assessment

The observation of effects on neurobehaviour was supported by changes in the expression of the three DNT markers *grin1a*, *dlg4*, and *gabra6a*, as observed earlier [15]. In contrast to that study, where all three target compounds altered the gene expression of these markers, we only observed upregulation of the expression of *gabra6a* by all three pharmaceuticals, whereas *dlg4* was only upregulated by CBZ. However, Thomas et al. carried out a gene expression study in the brain tissue of 75-day-old fathead minnows, whereas we investigated the entire zebrafish embryo at 5 dpf. Therefore, it is possible that the mRNA of other tissues in the embryo concealed the specific gene expression alteration in the brain. In zebrafish, *gabra6a* is mostly expressed in the photoreceptor cell layer of the retina and the cerebellum and can be detected only after 96 hpf [60]. The *gabra6a* gene encodes for the GABA receptor subunit 6α (Gabra6), which, together with subunit α4 and α5, plays a prominent role in GABA_α_ receptor function. Receptors containing these α-subunits are mostly extrasynaptic and mediate tonic inhibition [60]. Within the cerebellum, *gabra6a* expression occurs along with the development of GABAergic synapse formation, tonic conductance, motor control, and learning. Thus, even a temporal upregulation of *gabra6a* expression may lead to functional deficits of the GABAergic synapse formation in less active embryos [67]. Regarding the *dlg4* gene, it encodes for the postsynaptic density 95 (PSD-95) protein, which is a membrane-associated guanylate kinase (MAGUK) that contains multiple protein–protein interaction domains capable of inducing the clustering of postsynaptic receptors at excitatory synapses, such as the glutamatergic synapse [68,69]. From 72 hpf, PSD-95 protein is expressed in the zebrafish nervous system, specifically within the developing visual and olfactory system [69,70]. Since *dlg4* is strongly upregulated in the CBZ-exposed embryos, we may assume that the clustering of receptors at excitatory synapses is disturbed in the less active embryos.

### 4.5. Risk Assessment

**Table 5 ijerph-18-06717-t005:** Concentrations in surface water.

Location	CBZ	FLX	VNX	ref
NL	0.075–1.5 µg/L ^1^		0.02–0.22 µg/L	[71]
NL		0.004–0.75 µg/L ^2^		[72]
Germany	1.28 µg/L			[73]
Spain		0.12 μg/L		[74]
USA			1.31 μg/L	[75]
PNEC ^3^	1520 µg/L	10 µg/L	20 µg/L	

^1^ average-max; ^2^ min-max; ^3^ predicted no effect concentration= BMC_10_/ assessment factor 10, based on locomotion effects in ZFE.

Although the modes of action of the test pharmaceuticals may be comparable among vertebrate classes, the extrapolation of the observed effects in zebrafish embryos to humans requires additional investigation, e.g., regarding a human relevance analysis of the observed effects following a standardized protocol. The study compounds also affect locomotor activity in rodent models [76,77], though this has not (yet) been confirmed as a DNT effect. Further study is needed to assess whether the observed effects in the zebrafish embryos can be linked to irreversible effects in brain development and whether such effects on locomotor activity in zebrafish embryos are predictive for DNT-related locomotor effects in mammals. This in turn requires a definition of the applicability of the test in a toxicological pathway: for instance, in a validated adverse outcome pathway. Only then is a quantitative approach for human risk assessment justified. On the other hand, the test results in zebrafish embryos can be directly extrapolated to other fish species, and adversity can be suspected given the predicted impairment of behaviour in general, and more specifically, of vital functions such as feeding, fleeing, and reproduction. Therefore, the test results may directly feed into environmental risk assessment. Following the CRED protocol (criteria for reporting and evaluating ecotoxicity data [78]), BMC10 values are converted to predicted no-effect concentrations (PNEC) using a factor 10 (Table 5) and then compared to actual concentrations. In this way, it can be concluded from Table 5 that for the Dutch situation [71,72], the measured values for all three compounds are below the PNEC, although no more than a factor of 13.3 for FLX. However, given the of simultaneous presence of these (and other DNT-active compounds) in the environment, cumulative effects should be considered, and for cumulative risk assessment, EFSA suggested to apply dose addition as a default way to model mixture effects [79]. To test the validity of the dose-addition assumption for the compounds under study, the locomotor activity of the zebrafish embryos was also assessed after chronic exposure to the binary mixtures of the three pharmaceuticals. Our outcomes confirm dose addition as a consistent model for predicting the mixture responses, both in combinations of compounds of similar and dissimilar MOA. This confirms the emerging general concern regarding the daily human and environmental exposure to chemical combinations, although for this specific case, CBZ and VNX would not contribute substantially to the total of FLX units in a mixture dose-addition scenario, in view of their low potencies relative to FLX (see Table 2). In their studies, Thomas et al. (2012) and Kaushik et al. (2016) used environmental concentrations for different geographical regions, which were in a similar range to Dutch concentrations for CBZ and FLX, but higher for VNX (Table 5) [73,74,75]. Such a higher concentration of VNX, with an approximate potency of 10% compared to FLX (Table 2), would add substantially to the cumulative effect of combined FLX/VNX exposure and decrease the margin to the PNEC to a factor of about 10. Therefore, knowing that the numerous substances in surface waters can potentially contribute to the measured effect and that the possible combinations are more complex than the binary mixtures tested here, the added doses might easily reach an effective value. In addition, lower effective concentrations on neurobehaviour have been reported, including for FLX and VNX in ZFE [32,80], and for all three compounds in the predator escape test in 75-day-old fathead minnows [15,66]. It should, however, be noted that a direct comparison of effect concentrations across studies is often limited by differences in exposure route (e.g., bath exposure vs. microinjection). Therefore, differences in the outcome of studies should be regarded in the context of variations in study design, such as different sensitive windows, measured endpoints, analytical methods, and species.

Environmental chemical mixtures become increasingly concerning when one considers the added activity of the major metabolites which are commonly detected in the environment [81,82,83,84], sometimes at concentrations higher than the parent compounds. Additionally, some of these are equipotent to the parent compound, thereby strongly contributing to the overall effect, and as such, the observed inhibitory effect of norFLX and desVNX on zebrafish embryo locomotor activity supports the importance of including metabolites. Thus, the assessment of pollutant mixtures, including compound metabolites, is essential for understanding the overall DNT effects of environmental contaminants, in view of both human and environmental risk assessment, and the ZFE is a suitable model to screen for DNT effects.

## 5. Conclusions

Our study provides a new approach method characterized by a defined behavioural phenotype. The proposed model may be useful for the first screening of potential developmental neurotoxicants that can be used in the context of AOP-based screening approaches. Moreover, we proved the consistency of zebrafish embryos as a model for studying the combined effect of chemical mixtures in DNT. Future studies should further focus on the added value of the zebrafish behavioural test to predict DNT in humans, i.e., investigate additional behavioural endpoints reflective of neurodevelopmental disturbances, the irreversibility of the effects, and the relevance for adversity, all in the context of specific AOPs. The test is directly applicable in environmental risk assessment.

## Figures and Tables

**Figure 1 ijerph-18-06717-f001:**
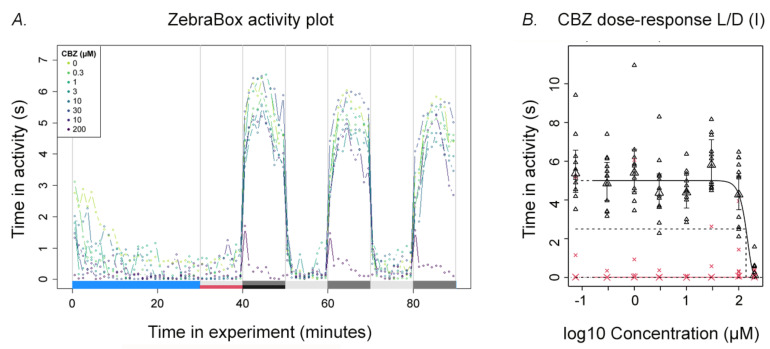
(**A**) A plot overview showing the entire light-dark transition test applied for the CBZ experiment, after 0–120-hpf exposure. The blue horizontal bar indicates the acclimatization period, while the grey and black bars represent the light and dark period, respectively. The X-axis shows the whole experiment time, whereas the Y-axis shows the time (s) spent in activity within 1 min by the zebrafish embryos (*n* = 12). Each dot shows the average time (s) spent in activity by 12 embryos/concentration in 1 min of recording, whereas different colours represent different concentrations shown in the upper left legend. (**B**) A dose–response curve of an individual CBZ experiment. The data set belongs to the first light-dark block of Figure 1A. The red crosses/lines represent the light period, whereas the black triangles/lines represent the dark period. The X-axis shows the CBZ dose range (µM) in log scale, whereas the Y-axis shows the time (s) spent in activity within 1 min. Each small symbol shows the time (s) spent in activity by each of the individual 12 replicate embryos within 1 min, while the large symbol represents the geometric means of *n = 12* together with their confidence intervals (error bars).

**Figure 2 ijerph-18-06717-f002:**
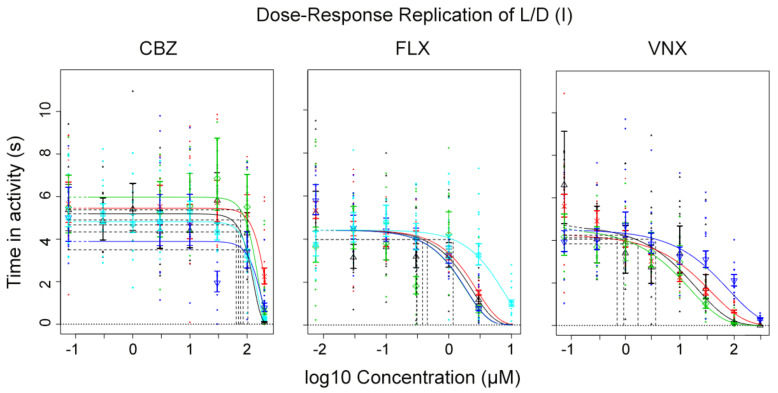
A dose–response curve plot showing five combined replicates of CBZ and FLX and four combined replicates of VNX (dark periods only). The individual experiments were used as covariate. One replicate of each compound was performed at VU, the others at RIVM. Symbols and error bars as in Figure 1B.

**Figure 3 ijerph-18-06717-f003:**
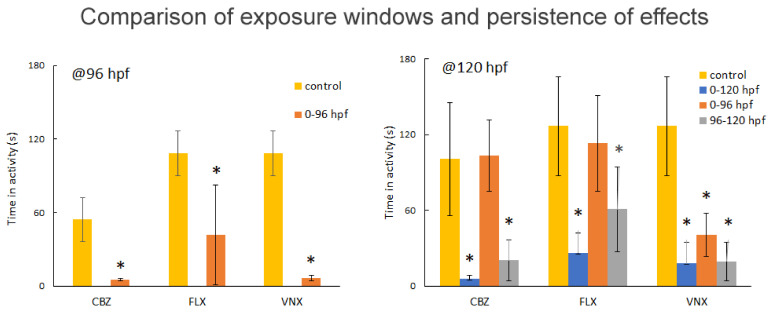
Persistence of effect on activity. Average total activity during the first 10’ dark period of *n* = 12 embryos, measured at 96 (left) and 120 (right) hpf, following exposure to 200 µM CBZ, 10µM FLX, and 300 µM VNX. The second and third dark block (not shown) provide identical results, whereas the light blocks (not shown) did not show statistical differences compared to the control. The activity (Y-axis) was measured as cumulative duration of movement (in seconds) during 10′. Bar colours indicate different exposure frames (see upper legend). The asterisk (*) indicates a significant difference compared to the control group (*p* < 0.05); error bars, SD.

**Figure 4 ijerph-18-06717-f004:**
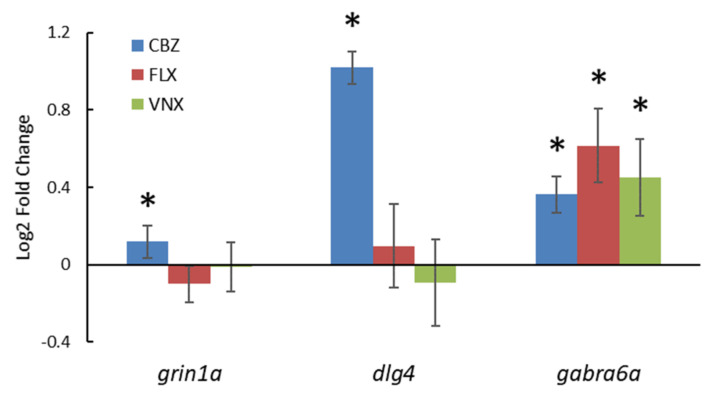
Effects of test compounds on marker gene expression. Gene expression analysis after 120 h of exposure to 115 µM CBZ, 107 µM VNX, and 6 µM FLX. The results are the average of *n* = 6 pools of embryos (each pool containing 12 embryos), expressed by using the delta–delta cycle threshold method (2^–∆∆Ct^). The asterisk (*) indicates significance, *p* < 0.05. Error bars, SD.

**Figure 5 ijerph-18-06717-f005:**
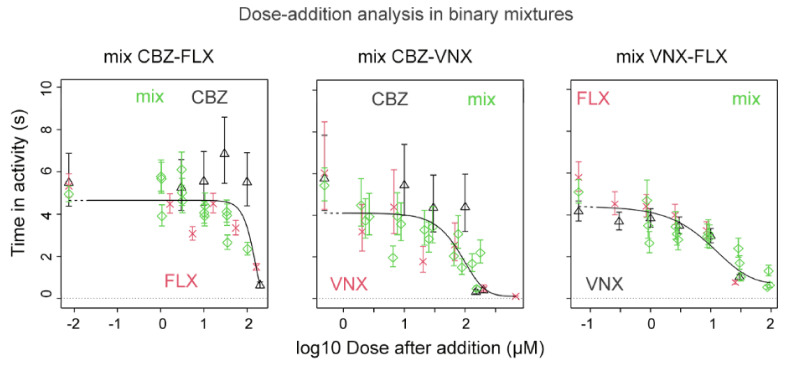
Dose–response mixture effects on swimming activity in zebrafish embryos upon exposure to a mixture of CBZ and FLX, CBZ and VNX, and VNX and FLX. The reference compound (black triangles/lines) is combined with the second compound (red crosses/lines); green diamonds/lines represent the corresponding mixtures. Doses of the second compound and of the mixtures are scaled to the reference compound. The mixtures do not deviate systematically from the overall fit, indicating that the combined effect results from dose addition.

**Table 1 ijerph-18-06717-t001:** Exposure dose range of the single-compound dose–response analysis at both experimental and environmental levels.

	Tested Dose Ranges (µM)	
	Experiments Environmental Levels	Dose–Response Experiments
Carbamazepine	0.0032 ^a^...100	0.3…200
Carbamazepine 10,11 -epoxide	-	0.03…30
Fluoxetine	0.00001…1	0.03…10
Norfluoxetine	-	0.03…10
Phenytoin	-	0.3…300
Venlafaxine	0.00032…100	0.3…300
Desvenlafaxine	-	0.3…300

^a^ Each series had intermediate half-logarithmic dilutions (min…max) and a blank control.

**Table 2 ijerph-18-06717-t002:** RPFs of the three compound combinations used for designing the mixture experiments.

Reference Compound	Second Compound	RPF *
Carbamazepine	Fluoxetine	50.50
Carbamazepine	Venlafaxine	12.65
Venlafaxine	Fluoxetine	10.26

* The RPFs were calculated using a specific function of PROAST for CBZ-FLX and VNX-FLX mixtures, whereas for CBZ-VNX mixture, it was obtained by comparing the BMC_50_ of the two compounds, using intermediate values from Table 3 (reference/second, i.e., second compound is factor x more potent than reference compound).

**Table 3 ijerph-18-06717-t003:** DR curves results of the single-compound analysis at experimental level, at 5% and 50% CES.

	BMC_05_ ^a^ (µM)	BMC_05_-CI ^b^ (µM)	BMC_05_ (mg/L)	BMC_05_-CI(mg/L)
Carbamazepine	51.55–82.25	38.8–103	19–12	9.16–24.33
Carbamazepine 10,11-epoxide	-	-	-	-
Fluoxetine	0.17–0.65	0.06–1.42	0.05–0.2	0.02–0.44
Norfluoxetine	2.21	0.2–6.03	0.65	0.06–1.78
Phenytoin	45.13	4.14–52.5	11.38	1.04–13.24
Venlafaxine	0.26–1.34	0.10–2.87	0.07–0.37	0.03–0.8
Desvenlafaxine	6.06	0.69–26.7	1.6	0.18–7.03
	BMC_50_ ^a^ (µM)	BMC_50_-CI ^b^ (µM)	BMC_50_ (mg/L)	BMC_50_-CI(mg/L)
Carbamazepine	116–185.1	103–216	27.4–43.73	19–51.03
Fluoxetine	1.41–5.40	1.01–6.77	0.44–1.67	0.31–2.09
Venlafaxine	9.53–48.45	6.19–63.9	2.64–13.44	1.72–17.73

-: No effect. ^a^ Benchmark concentration of the single-compound analysis calculated at the 5% and 50% effect level. The values represent the highest and the lowest BMC05/BMC50 of 5 (CBZ and FLX) and 4 (VNX) independent experiments in the exponential and Hill models. The experiments on the metabolites were performed once. ^b^ Confidence interval (CI) of the BMC5/BMC50. The values represent the highest BMCU05/BMCU50 and the lowest BMCL05/BMCL50 of 5 (CBZ and FLX) and 4 (VNX) independent experiments in the exponential and Hill models.

**Table 4 ijerph-18-06717-t004:** Quantitative evaluation of the dose addition: effect of mixtures on RPF of single compounds.

Reference Compound	Second Compound	RPF				
		Single Compounds		Compounds and Mix		
		Lowest	Highest	Lowest	Highest	Ratio of Overlap ^a^
Carbamazepine	Fluoxetine	51.7	63	47.5	61	0.85
Carbamazepine	Venlafaxine	4.26	5.91	5.11	9.3	0.86
Venlafaxine	Fluoxetine	9.67	13.1	5.25	12.9	0.75

^a^ Ratio of overlap was obtained by dividing the RPFL (relative potency factor lower) of the higher interval to the RPFU (relative potency factor upper) of the lower interval. This indicates an overlap of RPF confidence intervals without and with mixture. Ratio < 1 supports the hypothesis that the mixture effect can be predicted by dose addition.

## Data Availability

Not Applicable.

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
