# Peer review of "Developmental Neurotoxicity of Environmentally Relevant Pharmaceuticals and Mixtures Thereof in a Zebrafish Embryo Behavioural Test"

_ijerph, 2021, doi:10.3390/ijerph18136717_

Round 1
Reviewer 1 Report
Manuscript ID:N/A
Title: Developmental neurotoxicity of environmentally relevant pharmaceuticals and mixtures thereof in a zebrafish embryo behavioral test
General Comments:
The manuscript described the impact of common pharmaceutical products on the behavioral response and gene expression of developing zebrafish. The paper is clearly written and easy to follow, however the following major comments should be addressed prior to publication:
- The topic of this manuscript has been well studied and is not particularly novel to the field, there have been many studies of the effects of these pharmaceuticals on zebrafish and they need to be included in the intro. Particularly, some of the transgenerational work that has been done.
- The exposure concentrations are nominal and have not been analytically validated. This limits the usefulness of this study for risk assessment as studies based on nominal concentrations are often excluded from regulatory meta-analysis.
- The authors mention “environmentally relevant” in the title but do not specify how the exposure concentrations used relate to environmental relevance. Are they therapeutic doses? Doses found in waste water effluent? Authors need to provide more information.
Specific Comments:
Line 13: “limitedly performed” is awkwardly phrased. Consider revising.
Line 20: what is the difference between a behavioral response vs a “neurobehavior”? Aren’t all behavioral responses neurologically mediated?
Line 55: Citation is needed.
Line 96: Citation is needed.
Line 105: Including actual concentrations as opposed to just descriptors like “low” would be helpful in the intro of this study. What is considered a low vs a high concentration?
Line 128: This sentence is the exact same as a sentence earlier in the intro, this section of the manuscript requires significant revision prior to publication
Line 159: Why were two different strains of zebrafish used? Is the wild-type import essentially a pet-store strain?
Line 174: “comparable, although slightly different”… this is insufficient detail, and a brief description of the culture condition differences should be included in the manuscript.
Line 182: What were the 6-well plates made of? How was potential adsorption of the test compound accounted for?
Line 184: What is the composition of the embryo medium?
Line 214: How was the acclimation period chosen? Is 30’ 30 seconds or 30 minutes?
Figure 1 and 2- has very poor resolution and is difficult to interpret. Figures in a paper should look like the belong to the same study, some color theme/figure style should be used consistently throughout the manuscript.
Reviewer 2 Report
Subject: Manuscript submmited to International Journal of Environmental Research and Public Health, MDPI.
Developmental neurotoxicity of environmentally relevant pharmaceuticals and mixtures thereof in a zebrafish embryo behavioural test. Alessandro Atzei et al., 2021
In this manuscript the authors are presenting their research work using biological tests on Zebrafish embryos as a model, in which neurotoxicity tests were carried out. The use of Zebrafish embryos has been an excellent laboratory model for biological tests, which has the advantage of being cheaper, significantly reducing the use of superior vertebrate animals and their suffering throughout the experiments. In this work, the authors studied exposuring of the Psycho-pharmaceuticals substances such as carbamazepine (CBZ), fluoxetine (FLX), venlafaxine 24 (VNX), as well as some of its metabolites, on the neuronal embryonic development of zebrafish (Danio rerio). According to the authors, the data showed inhibitions of locomotor activities in cumulative exposure tests, indicating that the model is efficient to assay toxicity of samples. After careful reading of the manuscript, it is my opinion that the data are interesting and bring news in the area. Comments 1- Please change the definition of DNT (developmental neurotoxicity) from lines 15-16, to line 13, as this is the first time that the abbreviation appears in the text.
2- In lines 55-57, when the authors are commenting on tests currently used on animals to study DNT … they wrote ....However, the predictivity of these animal tests for human health effects is uncertain given model differences and also because of the relatively nonspecific or insensitive endpoints that are generally used to investigate DNT in animals. … Here it would be interesting for them to discuss why tests using zebrafish (Danio rerio) are more interesting in aspects criticized for other animals.
3- In line 59-60, the authors comment on the uses of in silico methods for DNT studies, but without indicating what these methods are, and their acceptance in the literature. They could briefly indicate these tests and their efficiencies in the area.
4- The data described in lines 96-99 are alarming, without considering other polluting agents such as pesticides, fertilizers and air pollutants. This can undoubtedly explain the high incidence of cancer that has been plaguing humans worldwide. In this part of the text, the authors could discuss something about the indication and use of the model also in tests of environmental pollution of water or animals exposed and used as food such as fish and seafood.
5- In the chapters of Materials and Methods and Results, in the section where the authors studied Zebrafish Embryotoxicity Test (ZFET), lines 175-198. The text is clear and easy to understand. However, the authors explain that the exposures to the different agents were for 72 hours. Any reason for this time? Why not an exposure in a time-dependent curve?
6- On lines 321-323, when the authors performed Zebrafish Embryotoxicity Test (ZFET) and showed that fluoxetine (FLX and its metabolite norFLX) resulted in a concentration-dependent decrease in GMS, with 322 a BMC05 of 31.2 (CI 7.22-57.2) µM and 32.19 (CI 14.9-59.7) µM (Fig S1). Except in the conclusion that there were signs of toxicity in the conditions used. How to extrapolate this data to the human model?
7- On lines 355-357, when the authors made Behaviour testing. Single compound dose-response analysis .... Exposure to the parent compounds, CBZ, FLX, and VNX was repeated at environmentally relevant concentrations, to confirm an absence of effects at these levels (Table 1). As in the title of the manuscript …. Developmental neurotoxicity of environmentally relevant pharmaceuticals and mixtures .... why were environmentally relevant concentrations studied?How to extrapolate these concentrations to clinical concentrations used routinely in humans? To really show a correlation between the toxicity tests done herein and a potential toxicity for humans, shouldn't the authors use plasma or blood concentrations achieved in dosages and used in chronic treatments? and then check for possible signs of toxicity in Zebrafish embryos?
8- How can the authors explain the differences obtained in the treatments of embryos with Carbamazepine in conditions without light and with light? (lines 413-420).
9-On lines 372-373 …the gene expression of the specific DNT 372 markers, grin1a, dlg4, and gabra6a, was studied. ….I have the impression that here the authors could indicate references on why they chose to study these genes, although they comment in other parts of the text.
10- In the experiments for evaluating the expressions of different genes, at lines 371-377, why did the authors not show the results of the RT PCR as an Electrophoresis of the comparative analyzes between the samples? This has been more common in the published literature. Also, why did they not show changes in the proteins of these genes to prove changes in the expressed products? For example, using a Western Blotting reaction, which is common in the literature, since the expression of genes in their respective mRNA does not mean translated protein.
11- In lines 476-478, when the authors discuss aspects of substances that are already known to be neurotoxic, why do they not indicate the names of the substances instead of generalizing? ...However, only just over ten substances, including some metals, inorganic compounds, organic solvents, and pesticides, are currently considered as the most important developmental neurotoxicants [39, 40].
12- The main conclusion of the results shown in this manuscript indicated by the authors is that almost all the tested substances (lines 488-489) showed toxicity in the experimental conditions used. ... All the studied compounds, except for CBZ-10,11-E, appeared to induce an inhibitory effect of the zebrafish embryo locomotor activity… However, from a clinical point of view, these drugs have been used for years, providing good results in the treatment of patients, without showing major disorders as side effects! How to extrapolate the results described here to the clinic? Although the method was able to indicate signs of toxicity for tested samples under the laboratory conditions used, how much can this be extrapolated to the clinic, keeping in mind the proportions of differences between the experimental model and humans ?
13- In lines 504-558, when the authors postulate theoretical explanations about the mechanisms of action of the toxicities for tested drugs, using the studied model, although these hypotheses are consistent with the literature, in my opinion experimental data are lacking so that such conclusions are finally proven.
14- When studying gene expression (lines 559-583), authors describe discrepancies between the data obtained herein, in the model they have been studied with Zebrafish, and other data in the literature (Thomas et al. (2012) [13]). Aren't these differences, points that indicate difficulties in extrapolating these toxicity data to more superior models such as humans for example? As they themselves indicate on lines 587-600 ...
15- However, in my opinion, the discussions on extrapolations of the experimental results obtained and described in this manuscript and risks of contamination of the environment, with possibilities of bioaccumulation in fish are more reasonable and can undoubtedly be extrapolated, since the models are similar. (lines 614-643).
In summary, in my opinion, a criticism that can be made in relation to these types of tests, using embryo of fish is the quantitative potential of extrapolating the results obtained. Is there a correlation between the concentrations used in experiments with zebrafish (Danio rerio) and the potential for injuries in humans? Especially in the therapeutic concentrations used? And the differences in the sizes of the models?
Round 2
Reviewer 1 Report
The authors have neglected to respond to some of my previous comments, for that reason I believe the revisions were insufficient.
Author Response
Please see the
Response to Reviewer 1 Comments
Point 1: The authors have neglected to respond to some of my previous comments, for that reason I believe the revisions were insufficient.
Response 1: The authors would like to state that the questions and critical appraisal of the manuscript by reviewer #1 are highly appreciated. Following the reviewer’s comment in response to the revised version of the manuscript we carefully re-examined the questions and our respective responses to find issues that could be considered as neglected issues. The main issue raised by the reviewer which may not have been addressed to a satisfying level was the first comment on missing literature references and references to transgenerational data. Indeed, literature references had been missed there, resulting in a reflection on the literature that was not up to date. To better address this issue and include the latest literature on the test compounds, part of the Introduction and part of the Discussion section have been rewritten. These sections are highlighted in the submitted revised version of the manuscript. Rewritten sections now read:
Introduction:
For CBZ, FLX and VNX effects in aquatic organisms, including zebrafish, have been reported, including in zebrafish. For CBZ it has been shown that it alters the zebrafish’s behavior in conjunction with histopathological changes in the brain, and lethality observed at high concentrations [27-29]. For FLX it has been shown that it alters zebrafish behavior, for instance reduced anxiety-related behavior [30, 31]. For VNX, recent studies have shown that VNX exposure starting at early stages can alter neurobehaviour at 5 dpf [32, 33]. In addition, for FLX and VNX also transgenerational changes in neurobehaviour have been reported ([34-36]
Discussion:
VNX is an SNRI with slightly different therapeutic uses that FLX. The pharmacological mechanism of action of VNX revolves around modulation of serotonergic as well as norepinephrine and dopamine re-uptake. This mechanism is shown to be responsible for the decreased behavioural responses as a result of VNX exposure in fish [32, 33, 64-66], and may be related to enhanced neurogenesis in the hypothalamus, dorsal thalamus and preoptic area [32]. This is considered to underlie the persistent developmental neurotoxicity of VNX. Nevertheless, for both VNX and FLX also additional sexe-specific transgenerational effects on neurobehaviour have been demonstrated, rooted in disruption of the cortisol stress axis [34-36]. VNX is a SNRI, and therefore slightly different therapeutic uses than FLX. In addition to the serotonergic modulation, VNX may act by blocking norepinephrine (NE) and dopamine (DA) reuptake. Decreased behavioural response by VNX may be due to changes in brain monoamine concentrations and changes were reported in the expression of 80 hpf zebrafish genes belonging to the serotonergic, noradrenergic, and dopaminergic pathways after chronic exposure to VNX. Despite that, the inhibitory effect on NE and DA occurs at concentrations much higher than those applied in the current study. Alternatively, altered activity levels in zebrafish embryos have been linked to changes in neurogenesis following contaminant exposure. VNX may disrupt the early developmental events of the zebrafish brain as proved by the enhanced neurogenesis in the hypothalamus, dorsal thalamus, and preoptic area. Our study outcomes are in agreement with this evidence. The lack of locomotor recovery observed in 5 dpf zebrafish exposure to VNX supports a DNT involvement for this compound, indeed. On the other hand, removal of pharmaceuticals at 4 dpf led to recovery of locomotor function in 5 dpf zebrafish exposed to CBZ and FLX., This thus either not providing indicates evidence forno persistent interference of these two compounds with development of the CNS or indicates that developmental stages up to four days do not present a sensitive window for CBZ- and FLX-induced DNT.at that stage. However, to better explore the role of CBZ and FLX in DNT, prolonged and continued locomotory tests are needed. For instance Indeed, in literature it is described that exposure to FLX between 4 and 6 dpf induced induced a significant swimming decreasedecrease in swimming activity, persistent up to 14 dpf, whereas such effectsthis effect werewas not observed iwithn exposures before 4 dpf [31].
In response to the reviewer’s remark on internal concentrations, a section of the Discussion has been further rewritten. This part now reads:
In addition, lower effective concentrations on neurobehaviour have been reported, including for FLX and VNX at environmental concentrations in ZFE [32, 80], and at 0.4, 0.03, and 0.16 µM for respectivelyfor all three compounds CBZ, FLX, and VNX in the predator escape test in 75 days old fathead minnows [15, 66]. It should however be noted that a direct comparison of effect concentrations across studies is often limited by differences in exposure route (e.g. bath exposure vs. microinjection). Therefore, This difference differences in the outcome of studies may be explained byshould be regarded in the context of variations in study design such as different sensitive window, measured endpoint, analytical method, and species, and the critical contribution of study conditions is supported by the absence of observed behavioural effects at environmental levels of the target compounds in a number of other studies with various fish species and amphibians, applying a variety of behavioural tests.
attachment.